# Plasma virome and the risk of blood-borne infection in persons with substance use disorder

Abraham J. Kandathil [1], Andrea L. Cox[1], Kimberly Page[2], David Mohr[3], Roham Razaghi[4], Khalil G. Ghanem[1], Susan A. Tuddenham[1], Yu-Hsiang Hsieh[5], Jennifer L. Evans[6], Kelly E. Coller[7], Winston Timp [1,4], David D. Celentano[8], Stuart C. Ray [1] & David L. Thomas[1]✉

There is an urgent need for innovative methods to reduce transmission of bloodborne pathogens like HIV and HCV among people who inject drugs (PWID). We investigate if PWID who acquire non-pathogenic bloodborne viruses like anelloviruses and pegiviruses might be at greater risk of acquiring a bloodborne pathogen. PWID who later acquire HCV accumulate more non-pathogenic viruses in plasma than matched controls who do not acquire HCV infection. Additionally, phylogenetic analysis of those non-pathogenic virus sequences reveals drug use networks. Here we find first in Baltimore and confirm in San Francisco that the accumulation of non-pathogenic viruses in PWID is a harbinger for subsequent acquisition of pathogenic viruses, knowledge that may guide the prioritization of the public health resources to combat HIV and HCV.

[1] Department of Medicine, Johns Hopkins University School of Medicine, Baltimore, MD, USA. [2] Department of Internal Medicine, University of New Mexico, Albuquerque, NM, USA. [3] Department of Genetic Medicine, Johns Hopkins University School of Medicine, Baltimore, MD, USA. [4] Department of Biomedical Engineering, Johns Hopkins University School of Medicine, Baltimore, MD, USA. [5] Department of Emergency Medicine, Johns Hopkins University School of Medicine, Baltimore, MD, USA. [6] Department of Medicine, University of California San Francisco, San Francisco, CA, USA. [7] Core Diagnostics, Abbott Laboratories, Abbott Park, IL, USA. [8] Department of Epidemiology, Johns Hopkins University Bloomberg School of Public Health, Baltimore, MD, USA. ✉email: dthomas@jhmi.edu

High hepatitis C virus (HCV) incidences among PWID are reported worldwide and threaten the United States Department of Health and Human Services' and World Health Organization's 2030 hepatitis elimination goal to *reduce* new infections by 90%[1]. Because of the opioid epidemic, the incidence of HCV infections per 100,000 in the United States more than tripled from 0.29 to 1.04 between 2010 and 2017, after a decade of steady decline[2–5]. Likewise, the opioid epidemic has spread HIV-1 including to populations not previously affected[6–8].

In addition to HIV, HCV, and other viral pathogens, there are other viruses found in blood that are not known to cause disease in humans (non-pathogenic viruses)[9–11]. For example, a DNA metagenomic study done on over 8000 individuals identified 19 human viruses in blood[10]. These included pathogens belonging to *Herpesviridae* and non-pathogenic *Anelloviridae* members[10]. Collectively, these viruses are referred to as the virome which also includes virus-derived elements in chromosomes and viruses that infect other types of organisms that inhabit humans[9].

The risk of transmission of bloodborne pathogens is typically measured by their incidences, while their spread is inferred by the phylogenetic relationships of their sequences. Transmission studies have shown that HCV precedes HIV infection in PWID, and hence new HCV infections serve as strong predictors of communities at risk for HIV[7,8]. We hypothesized that in that same way, the dynamics and abundance of non-pathogenic bloodborne viruses might serve as sentinels to predict the risk of acquiring pathogenic viruses such as HIV-1 and HCV and to reveal insights into transmission networks.

We have extensively characterized the blood virome of PWID[12–15]. After previously studying plasma from PWID in Baltimore, Maryland including using unbiased, metagenomic approaches, we focused on frequently observed non-pathogenic components of the DNA (alpha-, beta-, and gamma torque viruses) and RNA (pegivirus C and pegivirus H) virome in blood. In this study to test our hypothesis, we first assessed the plasma of PWID enrolled in a unique Baltimore cohort that recruited participants (see methods) when they were still HIV-1 and HCV uninfected and monitored them longitudinally for new bloodborne infections[16]. Our assessment revealed an expansion of the non-pathogenic virome prior to acquisition of HCV, which we also confirmed in a similar PWID cohort based in San Francisco. We also demonstrate the utility of non-pathogenic virome sequences in transmission studies.

## Results

**Expansion of non-pathogenic plasma virome prior to acquisition of HCV.** We studied 20 participants (Table 1) from a Baltimore based PWID cohort before and after HCV seroconversion ($HCV_{neg\ to\ pos}$ PWID). Participants were assessed at baseline, a median of 140 days (IQR:168.5) before their first seroconversion visit and then followed-up a median of 1184 days (IQR:213) after baseline assessment. Identification of non-pathogenic virome components was done using PCR (see methods). We similarly tested at two time points, 20 age- and gender-matched PWID (Table 1) from the same cohort who never acquired HCV or HIV ($HCV_{neg\ to\ neg}$ PWID) after a similar period of observation (median:1096 days, IQR:152.5). Consistent with our hypothesis, at baseline more non-pathogenic virome components were detected in those who would later acquire HCV than those who would not ($P = 0.0031$). The same assessments performed on plasma from 20 sociodemographic and age matched controls from urban Baltimore with no history of injection drug use revealed even fewer non-pathogenic viruses (Fig. 1a, $P = 0.0046$).

**Table 1 Subject characteristics.**

| | $HCV_{neg\ to\ pos}$ PWID ($n = 20$) | $HCV_{neg\ to\ neg}$ PWID ($n = 20$) | Non-injection drug users ($n = 20$) | $HCV_{neg\ to\ pos}$ PWID ($n = 19$) | $HCV_{neg\ to\ neg}$ PWID ($n = 19$) | $HCV_{neg\ to\ pos}$ PWID ($n = 13$) |
|---|---|---|---|---|---|---|
| City, Country | Baltimore, USA | Baltimore, USA | Baltimore, USA | San Francisco, USA | San Francisco, USA | Chang Mai, Thailand |
| Female (%) | 6 (30) | 6 (30) | 10 (50) | 4 (21) | 4 (21) | 2 (15) |
| Median (IQR) age years | 26 (6.25) | 27 (4) | 26.5 (6.77) | 26 (4) | 24 (4.5) | 23 (17) |
| No. of time points tested | 2 | 2 | 1 | 2 | 2 | 1 |
| Median (IQR) days between time points tested | 1184 (213) | 1096 (152.5) | NA | 427 (55) | 378 (34.5) | NA |
| Median (IQR) days before 1st HCV seroconversion visit | 140 (168.5) | NA | NA | 91 (32) | NA | ~180[a] |
| Median (IQR) duration in years of injection drug use | ≥6 months[b] | ≥6 months[b] | NA | 5 (3.76) | 3.97 (8.24) | NA |

$HCV_{neg\ to\ pos}$ PWID: Injection drug users who become HCV seropositive.
$HCV_{neg\ to\ neg}$ PWID: Injection drug users who remain HCV seronegative.
*IQR* interquartile range, *NA* not applicable.
[a]Samples were collected every 6 months.
[b]≥6 months was the inclusion criteria for the cohort.

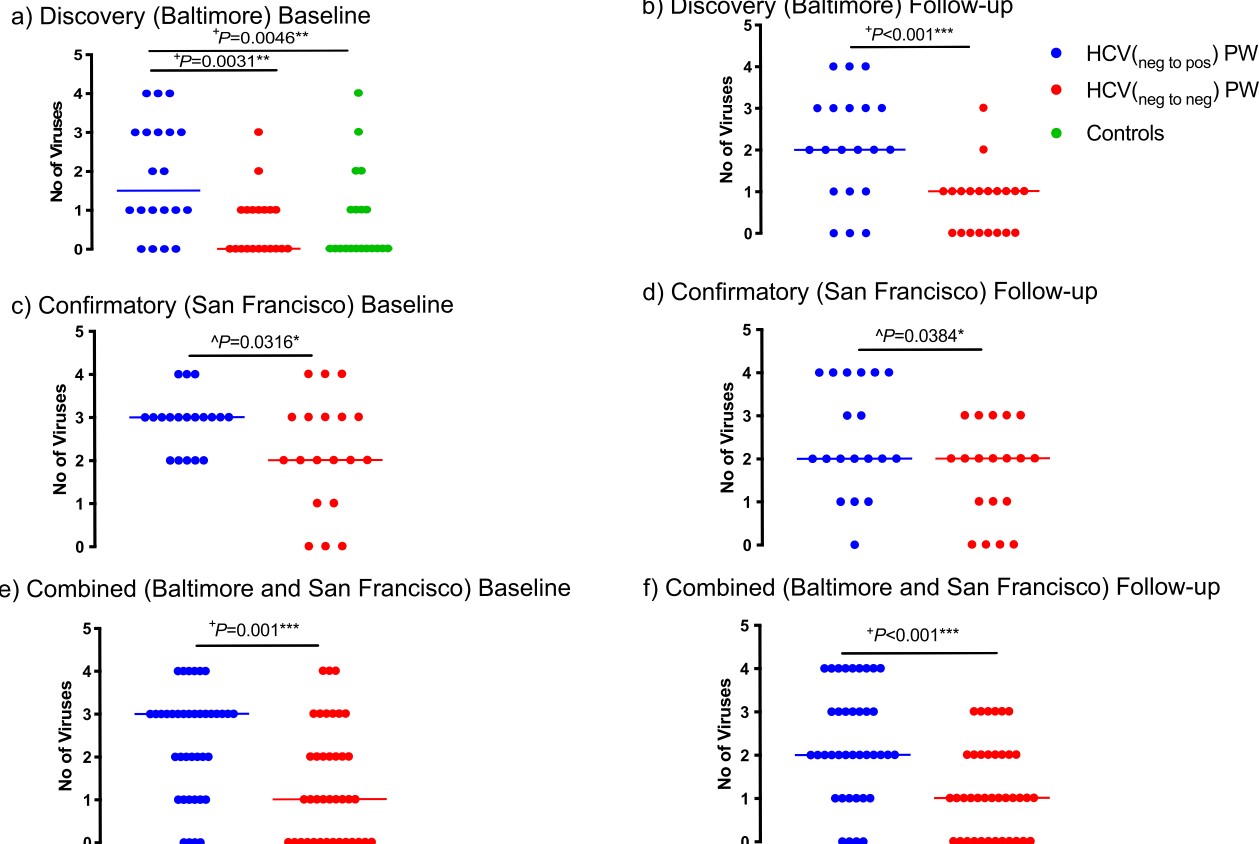

**Fig. 1 Non-pathogenic viruses accumulate more in PWID who later acquire HCV (HCV$_{neg\ to\ pos}$) than those who do not (HCV$_{neg\ to\ neg}$) and controls who do not inject drugs.** In the discovery (Baltimore) panel at baseline, greater non-pathogenic virome components were observed in HCV$_{neg\ to\ pos}$ PWID when compared to (**a**) HCV$_{neg\ to\ neg}$ PWID ($p = 0.0031$) and non-injection drug users (controls) ($p = 0.0046$) at baseline and to (**b**) HCV$_{neg\ to\ neg}$ PWID at follow-up ($p < 0.001$). Controls from Baltimore were composed of non-injection drug users tested only at one time point. At baseline, HCV$_{neg\ to\ pos}$ PWID were a median of 140 days before their first HCV seropositive visit. Each group was composed of 20 biologically independent specimens. (**c**) At baseline ($p = 0.0316$) and (**d**) follow-up ($p = 0.0384$), HCV$_{neg\ to\ pos}$ PWID had significantly greater virome richness in the confirmatory (San Francisco) panel (median:91 days prior to seroconversion) For panels **c** and **d** each group was composed of 19 biologically independent specimens. The differences were also observed on combining study participants ($n = 39$ biologically independent specimens in each group) from both panels **e** ($p = 0.001$) and **f** ($p < 0.001$). Error bar denotes median. $^{+}$Two tailed $P$ and $^{\wedge}$ one tailed $P$ value were calculated using Mann–Whitney or Kruskal–Wallis test. HCV was not included in the analysis. List of non-pathogenic viruses identified are shown in Supplementary Table 1. Source data are provided as a Source Data file.

At follow-up, HCV$_{neg\ to\ pos}$ PWID group had a median of two non-pathogenic viruses compared to one in the HCV$_{neg\ to\ neg}$ PWID group (Fig. 1b, $P < 0.001$).

We reasoned that additional pegivirus C infections not detected would have occurred and subsequently cleared before baseline, and hence also performed serological testing to assess for previous pegivirus C exposure[13,14]. By baseline, antibodies to pegivirus C were already detected in 35% and 30% among HCV$_{neg\ to\ pos}$ PWID and HCV$_{neg\ to\ neg}$ PWID, respectively. Accounting for resolved pegivirus C infection at baseline, further confirmed higher viral infections prior to HCV seroconversion in the HCV$_{neg\ to\ pos}$ PWID group compared to the HCV$_{neg\ to\ neg}$ PWID group ($P = 0.0041$). Pegivirus H antibodies were only detected in one sample from an HCV$_{neg\ to\ pos}$ PWID at follow-up. We also observed a greater circulating plasma DNA virome richness among HCV$_{neg\ to\ pos}$ PWID compared to HCV$_{neg\ to\ neg}$ PWID both at baseline ($P = 0.0038$) and follow-up ($P = 0.0016$). Similar differences were not observed in the RNA compartment.

To confirm these observations in the discovery panel (Baltimore) regarding expansion of the virome prior to acquisition of bloodborne viral infection among PWID, we performed the same assessments on a confirmatory panel composed of PWID from San Francisco[17]. Like the Baltimore cohort, this cohort also recruited participants (see methods) when they were HIV-1 and HCV uninfected and monitored them longitudinally for new bloodborne infections. We studied 19 age and gender-matched participants from the HCV$_{neg\ to\ pos}$ PWID and HCV$_{neg\ to\ neg}$ PWID group at two time-points (Table 1). The HCV$_{neg\ to\ pos}$ PWID were tested a median of 91 days (baseline, IQR:32) prior to HCV seroconversion. As in the discovery panel (Baltimore), we detected a greater expansion of the plasma virome among HCV$_{neg\ to\ pos}$ PWID compared to HCV$_{neg\ to\ neg}$ PWID in the confirmatory panel (San Francisco) (Fig. 1c, d). Thus, in these two independent cohorts, we detected and then confirmed our inference that the risk of a bloodborne viral pathogen acquisition is preceded by expansion of the plasma virome ($P = 0.001$, Fig. 1e) that is maintained at follow-up ($P < 0.001$, Fig. 1f). Further evidence of high burden of viral infections among PWID prior to HCV seroconversion was observed when we tested plasma from 13 members of yet another cohort of HIV negative PWID from Chang Mai, Thailand (Table 1) prior to HCV seroconversion (Supplementary Fig. 1)[18]. Thus, the accumulation of non-pathogenic viruses (Supplementary Table 1) in plasma of PWID identifies those who are at risk for medically important viruses like HCV and HIV.

**Utility of non-pathogenic plasma virome sequences in identification of drug use networks**. It can be important to understand the structure of the drug-use networks that transmit bloodborne infections. Yet, limitations in ascertainment of personal, illegal practices leave most transmission events uncharacterized even with post-infection analysis of HCV sequences. We asked if the phylogenetic relationships can be revealed by sequencing the non-pathogenic virome components. We initially performed Sanger sequencing on Baltimore and Chang Mai samples with pegivirus C infection, including one transmission pair, containing pegivirus C RNA and HCV RNA (Supplementary Fig. 2). As expected, sequences from Baltimore clustered distinctly from those recovered from Thailand, and pegivirus C sequences from persons in known HCV-defined groups also clustered as did samples from same study participants at first and second time points (except when the initial virus sequence was cleared). In addition, pegivirus C sequences revealed new linkages among PWID previously not recognized by HCV sequences and/or self-report.

Based on our observation of the plasma DNA virome, we characterized the predominant non-pathogenic components of the DNA virome, *Anelloviridae*. Seven plasma samples from four HCV-infected PWID partnerships identified by epidemiological data with *Anelloviridae* viremia were subjected to next generation sequencing (NGS) using nanopore (see methods). Following NGS, reads were taxonomically classified using Centrifuge[19], a microbial classification engine, that is part of 'What's in my Pot' workflow used for real-time analysis of nanopore sequences. Classified sequences were then used for construction of full-length de novo assemblies using Canu (v1.9)[20]. To independently validate multi-species infection, Sanger sequencing was done to confirm the presence of four distinct alphatorquevirus species identified in a study participant. Reads classified as TTV7, TTV8, TTV9, and TTV18 were used to design PCR primers targeting a 500 bp region of ORF1 to confirm their presence in the plasma. The ORF-1 region was targeted since ICTV guidelines use a cut-off value of 35% nucleotide sequence identity in the ORF-1 as a demarcation criterion for species identification.

We then designed primers based on reads classified as TTV8 to amplify positions 275–400 of ORF-1[21]. This sequence contained a relatively conserved region of ORF-1 in the 5' end followed by the first 100 amino acids of the HVR-1 region located in ORF-1[22]. For a single participant, the same set of primers were also used to detect TTV8 at time point 29 months later. Using study participant sequences ($n = 6$) and four non-Baltimore sequences that included ICTV reference sequences for TTV7 and TTV8, amino acid sequence divergence was calculated using Mega X[23] (Fig. 2). Pairwise distance confirmed relatedness of alphatorquevirus sequences observed between a drug use pair previously identified using HCV core-E1 sequences. In addition, we also observed drug use pairs linked based on alphatorquevirus sequences but not identified using HCV sequences, suggesting that alphatorquevirus sequences might reveal information about networks prior to HCV infection and unique from HCV sequences.

## Discussion

These data confirm that the blood virome can reveal the risk of bloodborne infectious diseases and that the phylodynamic structure of the virome may provide unique insights previously hidden among PWID.

Characterization of non-pathogenic components of the virome has clear advantages over traditional epidemiological approaches of viral pathogen sequencing. By using non-pathogenic viruses as sentinels, risk can be ascertained before harm occurs in a population. In principle, this is similar to the use of HCV as a marker of HIV risk. HCV infection was widespread in Scott County Indiana before HIV was introduced and early recognition of HCV might have prevented at least 58 cases of HIV[7,8]. Since HCV causes more mortality in the United States than HIV, HCV is clearly a suboptimal sentinel[24,25]. Since we found that non-pathogenic members of the virome were evident before HCV, this approach might be more effective. In addition, identification of sequences from multiple anellovirus species in study participants using nanopore suggests that the PCR based approach likely underestimated the number of members among individual study participants.

There are also important limitations to existing methods of identifying the sources of bloodborne pathogen transmission. Because of stigma and legal issues, self-identification of networks is an insensitive method to detect transmission pairs. Even phylogenetic approaches focused on one virus have rarely been helpful to reveal drug use networks. With HCV, natural clearance, reinfection, superinfection and within-host evolution may render sequences obtained at a particular point in time uninterpretable[26]. The addition of even one more virus might exponentially increase the probability of phylodynamic inferences. For example, if an analysis of HCV sequences produced a probability of a shared source of 0.90 and the same pair also shared *Anelloviridae* sequences that suggested the same probability, then the actual probability of sharing rises from 0.90 to 0.99. While this example is simplistic and the assigned probabilities arbitrary, next generation sequencing has already been proposed for the Advanced Molecular Detection Program of the United States Centers for Disease Control and Prevention[27].

Our data reveal the vast potential applications of full characterization of the blood virome that could transform our understanding of PWID networks at a crucial time when the opioid epidemic is spreading worldwide. While this work provides a virologic foundation, additional work is needed to increase non-pathogenic virome sequences to aid taxonomic identification and for development of suitable phylodynamic methods

## Methods

**Study participants**. Study samples were drawn from a cohort of injection drug users located in Baltimore, Maryland (USA)[16]. At the time of recruitment, participants in the discovery panel (Baltimore) reported injecting drugs in the last 6 months and were negative for both HIV and HCV infections. Participants are then longitudinally followed with plasma and serum collected at each monthly visit. At each visit, participants are tested for HCV and are tested for HIV every 6 months. For this study, we identified 20 participants in whom HCV seroconversion occurred (HCV$_{neg\ to\ pos}$) and there was sufficient stored plasma from before and after HCV infection. The initial HCV negative sample was a control for the ultimate acquisition of HCV. In addition, from the same cohort a total of 20 additional controls were identified with similar time of follow-up but who did *not* acquire HCV infection (HCV$_{neg\ to\ neg}$). Participants from the Baltimore cohort received a participant compensation of US$25 for all study visits.

All 40 participants from the HCV$_{neg\ to\ pos}$ PWID and HCV$_{neg\ to\ neg}$ PWID groups were tested at two-time points, separated by a median of 1112.5 days. For the HCV$_{neg\ to\ pos}$ PWID, the first time point was a median of 126 days before seroconversion. The median age in the HCV$_{neg\ to\ pos}$ PWID group was 26 years (IQR:6.25) and in the HCV$_{neg\ to\ neg}$ PWID group it was 27 years (IQR:4). Each group had a gender ratio of 2.33:1. For the four transmission pairs, each pair was tested at two visits that were a median of 1085 days (IQR:180) apart.

Since so little was known about the plasma virome of young adults, we also added sociodemographic controls who denied injection drug use:10 men from the Johns Hopkins Emergency Department[28] and 10 women from the gynecological clinic[29]. The control group had a median age of 26.5 years (IQR:6.77) and a gender ratio of 1:1. Finally, for confirmation, a geographically distinct cohort of HIV negative PWID from San Francisco, California (USA)[17] was used. Similar to the discovery panel, participants in this confirmatory panel reported injecting drugs in the last 6 months and were negative for both HIV and HCV infections. We included 19 HCV$_{neg\ to\ pos}$ PWID and HCV$_{neg\ to\ neg}$ participants. Each of the total 38 participants were tested at two timepoints. The median age in the HCV$_{neg\ to\ pos}$ PWID group was 26 years (IQR:4) and in the HCV$_{neg\ to\ neg}$ PWID group it was 24 years (IQR:4.5). Each group had a gender ratio 3.75:1. HCV$_{neg\ to\ pos}$ were tested a median of 91 days (IQR:32) before HCV seroconversion and a median of 427 days

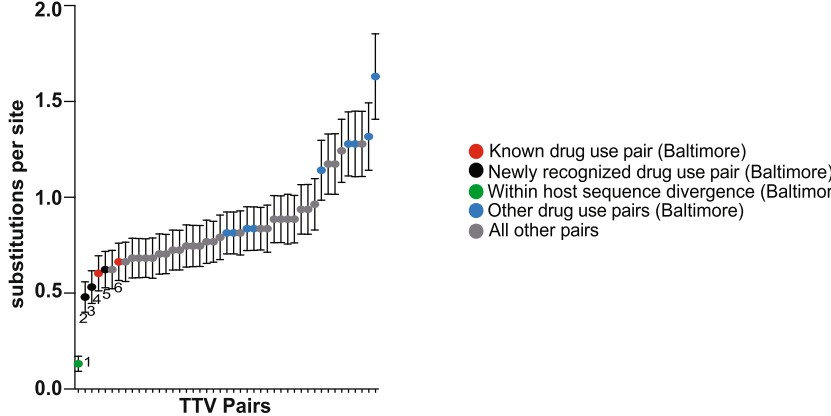

| TTV Pair | Pair Subject IDs | Time between sample collection (months) |
|---|---|---|
| 1 | Subject A(v1) − Subject A(v2) | 29 |
| 2 | Subject A(v1) − Subject D | 8 |
| 3 | Subject A(v2) − Subject D | 28 |
| 4 | Subject B − Subject A(v1) | 5 |
| 5 | Subject B − Subject D | 13 |
| 6 | Subject B − Subject A(v2) | 41 |

**Fig. 2 Amino acid sequence divergence between alphatorquevirus pairs confirms known drug use pair while also revealing newly recognized drug use pairs.** The number of amino acid substitutions per site between alphatorquevirus ORF-1 (positions: 263–404) sequences are shown. Error bar denotes standard error estimate obtained by a bootstrap procedure (1000 replicates). Analyses were conducted using the Poisson correction model. This analysis involved 10 amino acid (6 Baltimore and 4 non-Baltimore) sequences from biologically independent samples and 97 positions. All positions containing gaps and missing data were eliminated. Evolutionary analyses was conducted in MEGA X. Table shows drug use pairs identified using alphatorquevirus sequences and their temporal relation. (Gen Bank ID: AB054648, JN980171, AB054647, AF261761, MZ889122-MZ889127). All other pairs denote sequence divergence between Baltimore and non-Baltimore alphatorquevirus.

(IQR:55) after their first visit. The HCV$_{neg\ to\ neg}$ were tested at two timepoints a median of 378 days (IQR:34.5) apart. UFO participants were remunerated for all study visits including HCV screening (US$10) and follow up visits (US$20-$25). Also included were 13 HCV$_{neg\ to\ pos}$ PWID from Chang Mai, Thailand 6 months before seroconversion[18]. The Thai cohort had a median age was 23 years (IQR:17) and a gender ratio of 5.50:1.

Informed consent was obtained from all participants in all studies prior to sample collection. For persons studied from cohorts that are not active (that is, all but the Baltimore and San Francisco PWID group), as an additional precaution, an identity unlinked procedure was implemented prior to any testing. This and all other study procedures were approved by the Johns Hopkins Institutional Review Board.

**Molecular detection and serology.** Extraction of plasma RNA and DNA was as previously described[12]. Briefly, the *Quick*- DNA/RNA miniprep plus kit (Zymo Research, Cat# D7003) was used to extract DNA and RNA from 200 uL of plasma. Pre-extraction steps included spinning the samples at 1600 g for 15 min at 4 °C to remove debris (e.g., insoluble complexes) followed by filtration using 0.2 μM syringe filters (Fisher Scientific, Cat# 6778-1302). Following cDNA synthesis using superscript IV (Thermo Fisher Scientific Cat#18090010), pegivirus H infection was detected using primers described by Berg et al. (Supplementary Table 2)[30]. The single round PCR targeted a 160 bp region in the NS2-3 junction of pegivirus H. Pegivirus C detection was done using a one-step SuperScript™ IV One-Step RT-PCR (Thermo Fisher Scientific Cat#12594100) protocol followed by second round PCR using Platinum Taq DNA polymerase High Fidelity (Thermo Fisher Scientific Cat#11304-011). A hemi-nested PCR targeted nucleotide positions 979–1650 in the E2 region of pegivirus C as previously published[31]. The first step used primers E2_FP and E2_ORP while the second round used E2_FP and E2_IRP (Supplementary Table 2). Sequencing of HCV core-E1 (nucleotide positions: 843–1316) for the samples was also done as previously described (Supplementary Table 2)[32]. The nested PCR used outer primers CE1_ESP and CE1_EAP and inner primers CE1_ISP and CE1_IAP.

Previously described primers were used to detect infection with alpa-, beta-, and gamma- torquevirus (Supplementary Table 2)[33]. The PCR amplicons facilitated identification and differentiation of infections due to alpha-, beta-, and gamma torquevirus. The first round PCR primers amplified a conserved region found in the 5' end of all three torquevirus genera using forward primers NG779/NG780 and reverse primers NG781/NG782. Genus specific primers for alpha-(NG779/

NG780 and NG785), beta- (NG792/NG793/NG794 and NG791), and gamma (NG795 and NG796) torqueviruses allowed for differential detection of each genus. The size of the resulting amplicons was used to differentiate the three genera – alpha- (112–117 bp), beta- (70–72 bp), and gamma- (88 bp) torquevirus. The PCR for alpha torquevirus detection was semi-nested in design while a nested PCR for beta- and gamma torquevirus was used. Additionally, we enhanced sensitivity of the PCR by incorporating a rolling circle amplification (RCA) based preamplification step. We also observed increased sensitivity of RCA + PCR without the initial 95 °C denaturation step for RCA. Enrichment of circular templates using RCA was done using TempliPhi™ (Sigma Aldrich GE25-6400-10). RCA was carried out at 30 °C for 18 h with 2 uL of DNA as input. For the PCR, a 1:5 dilution of RCA concatemers was taken for subsequent PCR reactions.

Serology testing was performed to detect IgG antibodies specific for pegivirus H and pegivirus C on the ARCHITECT immunoassay platform using previously described antibody assays[34]. Briefly, separate anti-IgG assays were developed that use pegivirus H recombinant antigens, NS4AB or E2, or pegivirus C recombinant antigen, E2, to detect IgG specific responses. Provisional assay cut-offs were based on screening a volunteer donor population that was negative for pegivirus H and pegivirus C RNA or antibodies. A signal to cut-off (S/CO) value greater than or equal to 1.0 was considered reactive in the pegivirus H or pegivirus C assays.

**Nanopore sequencing.** Sequencing of *Anelloviridae* was done using Oxford Nanopore Sequencing on the GridION platform. The concatemers generated using RCA were digested using T7 endonuclease to debranch and generate fragments of varying sizes. The fragmented DNA was subjected to a magnetic bead-based size exclusion cleanup protocol to keep fragments size between 500 and 10,000 bp. Following the size based clean up, samples that had a concentration greater than 500 ng were selected for library preparation using the using SQK-LSK109 kit with native barcoding EXP-NBD104. Libraries were pooled in equimolar amounts and loaded onto a single R9.4 flowcell. Sequencing and basecalling was performed on the GridION running 19.12.2 according to protocol.

The reads were analyzed using What's In My Pot (WIMP), a quantitative analysis tool for nanopore reads[19]. As part of the WIMP workflow, taxonomic classification of the reads was done using Centrifuge[19]. For classification of reads, Centrifuge uses a database generated from sequences in RefSeq[35]. Classified reads for each alphatorquevirus species were used with Canu v1.9[20] (parameters: genomeSize=3k and -nanopore-raw) to assemble species-specific viral genomes. When primers were needed for Sanger sequencing of specific alphatorquevirus

sequences, samples were aligned to reference ICTV species, indexed used SAM tools[36], and visualized on integrative genomics viewer[37]. Sanger sequencing was used to confirm the presence of multiple alphatorquevirus sequences in a single study participant and to obtain sequences for phylogenetic analysis.

**Statistical analysis**. Since data were not normally distributed, nonparametric statistical tests were used to characterize the probability of the data were explained by the null hypothesis, that there was no difference in the viromes of persons who acquired HCV or did not. The number tested was determined to provide more than 90% power to detect a difference of two additional viruses in the virome of the $HCV_{neg\ to\ pos}$ PWID compared to $HCV_{neg\ to\ neg}$ PWID, and then by specimen availability. All statistical calculations were done using GraphPad Prism version 9.0.2 for Mac.

**Phylogenetic analysis**. Mega X was used for phylogenetic analysis of sequences[23]. To determine amino acid sequence divergence, amino acid sequence alignments were done on 10 alphatorquevirus ORF-1 (amino acid position: 263–404) sequences using the MUSCLE algorithm followed by analysis using Poisson correction model[38] with a bootstrap value of 1000. Six Baltimore alphatorquevirus sequences classified as TTV7 and TTV8 from PWID were used (GenBank ID: MZ889122-MZ889127) and four full-length non-Baltimore alphatorquevirus sequences were obtained from GenBank. This included TTV7 (AF261761) and TTV8 (AB054647) ICTV reference sequences. In addition, we also included all full length alphatorquevirus sequences (AB054648, JN980171) obtained when doing BLAST search with AB054647 (TTV8 reference sequence).

Phylogenetic relationships of HCV and pegivirus C nucleotide sequences was inferred using the maximum likelihood method based on the Tamura Nei model and a bootstrap value of 1000. A total of 34 pegivirus C (GenBank ID: MN938857-MN938890) envelope sequences obtained from PWID in Chang Mai (Thailand) and Baltimore (USA) were used. Similarly, 43 HCV (GenBank, ID: MN954475-MN954517) core-E1 sequences obtained from Chang Mai (Thailand) and Baltimore (USA) were used.

**Reporting summary**. Further information on research design is available in the Nature Research Reporting Summary linked to this article.

## Data availability
The sequence data generated in this study have been deposited in the GenBank database under accession code MN938857-MN938890 (pegivirus- C), MN954475-MN954517 (HCV), and MZ889122-MZ889127 (alphatorquevirus). The metagenomic dataset generated and analyzed during the current study have been deposited in Sequence Read Archive with the accession code PRJNA764703. The virome data used in Fig. 1 are provided in the Supplementary Information/Source Data file. Source data are provided with this paper.

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

## Acknowledgements
This work was supported in part by National Institute on Drug Abuse (NIDA) (R01DA01380 for D.L.T., and A.J.K., R21DA053145 for A.J.K., R01DA16017 for K.A.P.) and National Institute of Allergy and Infectious Diseases (NIAID) (U19AI159822 for A.L.C.). The funders had no role in study design, data analysis, decision to publish, or preparation of the manuscript.

## Author contributions

D.L.T. and A.J.K.: conception and design. A.J.K. and J.L.E.: collection and assembly of data. A.J.K., R.R., W.T., and D.M.: computational analysis. A.L.C., K.A.P., K.G.G., S.A.T., D.D.C., Y.H., and K.E.C. provision of the samples and study materials. D.L.T., A.J.K., and S.C.R.: analysis and interpretation of data. A.J.K. and D.L.T.: drafting of the article that was revied and edited by all the authors.

## Competing interests

S.T. has been a consultant for Biofire Diagnostics, Roche Molecular Diagnostics and Luca Biologics, and has received speaker honoraria from Roche Molecular Diagnostics. All other authors declare no competing interests.
