## [Peer Review File · Nature Communications]

Plasma virome and the risk of blood-borne infection in persons with substance use disorderReviewers' Comments:

Reviewer #1:

Remarks to the Author:

The manuscript by Kandathil et al describes an interesting set of experiments reporting more "orphan" viruses in PWID who have or will soon sero-convert to HCV than those who don't. Matched non-PWID show even fewer orphan viruses. The number of orphan viruses can therefore be used as a predictor of who is most likely to sero-convert to HCV. Presumably interventions could be targeted to PWID exhibiting more diverse virome and therefore are more exposed through contaminated needle sharing. The study samples were carefully chosen and age and gender matched. The results are definitely interesting but two technical points remain unclear to this reviewer.

1. How viruses were counted.

Anelloviruses make up the vast majority of the orphan viruses. It is first stated that previously described primers are used. Are these different primers for the alpha, beta, and gamma genera? Single, hemi, or nested PCR? Mention is then made of nanopore sequencing of the concatamers. Are these the RCA products or the PCR products (ligated?). How was sequence data used to enumerate the number of distinct anelloviruses? The following sentence is hard to follow: "Classified reads for each TTV were used with Canu (v1.9, parameters:genomeSize=3k and -nanopore-raw) to assemble species-specific viral genomes. corresponding Reference anelloviridae sequences obtained from the ICTV database (<https://talk.ictvonline.org/>)." Some samples yielded up to four anelloviruses. Are these different species of anelloviruses? Were the nanopore sequence reads mapped against the ICTV species to count them? Was ORF1 used or any part of the genome? Was there a minimum % of sequence difference used to count different viruses or species? What was done with the alpha, beta, gamma PCR products? How big were these anellovirus amplicons?

Were infections with multiple pegivirus C ever detected?

PCRs were used for the two pegiviruses but not clear if the pegivirus-H used a nested or hemi-nested PCR (ie primers by Collins et al4).

Was bleed over, the phenomenon observed with Illumina sequencing whereby a small fraction of reads from one sample end up with barcodes from another sample considered? Does it occur with nanopore?

The part of the material and methods, describing how viruses were counted, needs clarification.

2. Phylogenetic analyses.

Supp fig 2. The two trees are hard to follow and could benefit from more brackets or arrows to point out the relevant taxa. The HCV black taxa are clearly closely related but are readers supposed to interpret the other red circles that are closely linked as transmission pairs? Are these known pairs (or triplet in one case) confirmed by the PWIDs actually sharing injection equipment? Is the black HCV pair interesting because that relation between PWIDs was not known? Is the (weaker) linkage of the black pegiviruses used to confirm that these two PWID were directly connected (rather than there being an unknown intermediate PWID)? Non Baltimore pegi-C viruses are also clearly related. Are these longitudinal samples? From known contacts? Not clear what the point of all these taxa is? Is the point that there are also other pairs of very closely related pegiviruses among the non-Baltimore subjects? These tree need to be better explained and labeled.

The smaller TTV and HCV trees are also hard to comprehend at a glance. Is it the contention that TTV A and B are from a recent common ancestor (ie linked)? That subject B, C, and possibly D are also connected? That all the HCV taxa are closely connected? The very short length of virus sequenced

(111 nucleotides for TTV and 337 for HCV) typically make such phylogenetic linkage rather weak. Full genome sequencing usually used for such transmission linkages.

In fig 1 and S1 the pegiviruses cannot be differentiated from the anelloviruses. Are anelloviruses more likely than pegiviruses to be predictor of future HCV infection? Can anelloviruses alone do the job?

Also changing the term orphan viruses should be considered as it is not well known and a bit outdated. I suggest commensal viruses, non-pathogenic virome, or simply anelloviruses and pegiviruses. In text pegivirus C should be named as such not as Pegi-C.

Reviewer #2:

Remarks to the Author:

This manuscript by Kandathil et al tried to address the virome in plasma in persons with illegal substance abuse, and connect that to the transmission of important pathogenic viruses including HCV. This work in its current form is not yet adequate to address important medical or epidemiological questions, and also the content is short from substantially interesting/convincing for nature communications.

The hypothesis on virome difference between persons with drug usage and non-users is correct yet not totally unexpected, shared needles as well as other paths surely will increase the amount and diversity of viruses in drug users, plus the immunological side effects of many illegal substances. Will these changes lead to more susceptibility to TTV or HCV? Maybe, but it's not tested in this manuscript. Plus, what about direct transfer of pathogenic viruses?

In addition, it is quite unusual to see a manuscript NC without sections and only with two figures, this rather suggests insufficient work has been done on the research topic.

REVIEWER COMMENTS

Reviewer #1:

1) Anelloviruses make up the vast majority of the orphan viruses. It is first stated that previously described primers are used. Are these different primers for the alpha, beta, and gamma genera? Single, hemi, or nested PCR?

We apologize for poor elucidation of the methodology used to detect and enumerate the viruses. The methodology has been rewritten to improve the clarity of the molecular techniques used. To respond directly, for the first part of the study, a nested PCR (Ninomiya et al) was used to determine the dynamics and abundance of anelloviruses in people who inject drugs (PWID). The nested PCR amplicons aid in identification and differentiation of infections due to alpha-, beta-, and gamma torqueviruses. The first round PCR primers amplify a conserved region found in the 5' end of all three torquevirus genera. The second round used specific primers for differential detection of each genus. The size of the resulting amplicons was used to differentiate the three genera – alpha- (112-117 bp), beta- (70-72 bp), and gamma- (88 bp) torquevirus. The PCR for alpha torquevirus detection was semi-nested in design while a nested PCR for beta and gamma torque viruses was used. Additionally, we enhanced sensitivity of the PCR by incorporating a rolling circle amplification (RCA) based preamplification step (Fig 1 A&B). We also observed increased sensitivity of RCA+PCR without the initial 95° C denaturation step for RCA (Fig 2). The assumption was that removing the denaturation step would allow for an amplification bias towards single stranded circular genomes. Further standardization of the RCA+PCR also allowed us to use 2uL of extracted DNA as input (Fig 2). To summarize, a nested PCR using genus specific primers to differentiate alpha-, beta-, and gamma

torque virus infection was used to determine the dynamics and abundance of anelloviruses in PWID.

Figure 1. Preamplification using rolling circle amplification was found to increase PCR sensitivity

A) PCR only

B) RCA+PCR

Sensitivity of anellovirus PCR (A) was enhanced following incorporation of a rolling circle amplification step prior to PCR as shown in 1B. The figure shows four samples tested in duplicates - 1&2, 3&4, 5&6, 7&8. MW: molecular weight ladder

Figure 2: Modified rolling circle amplification without initial denaturation was observed to increase PCR sensitivity across a wide range of DNA input

Denaturation by heating to 95° C (Row1) prior to rolling circle amplification leads to lower efficiency of preamplification when compared to RCA without denaturation (Row2). MW: Molecular wt ladder; NC: Negative Control. The DNA input for rolling circle amplification for lanes 1-5 was 0.5 μL, lanes 6-10 was 1 μL, and for lanes 11-15 was 2 μL. A range of DNA dilutions were also used: 1:10 dil(Lanes 1,6,11), 1:100 dil (Lanes 2,7,12) and 1: 1:100 (Lane 3,8,13). Positive Controls: Lanes 4,9, and 14.

2) Mention is then made of nanopore sequencing of the concatamers. Are these the RCA products or the PCR products (ligated?). How was sequence data used to enumerate the number of distinct anelloviruses?

The next step after observing the dynamics and abundance of anelloviruses in PWID was to determine utility of their sequences in phylodynamic studies. The conserved short sequences of

the nested PCR amplicons precluded their use in phylogenetic studies. In addition, high sequence diversity coupled with a dearth of data on prevalent anellovirus species among PWID made it difficult to design suitable PCR primers. Hence, an unbiased approach using metagenomic sequencing was incorporated to uncover anellovirus sequences. Considering our need to for long sequencing reads to help in species level identification, the nanopore was used to generate sequence data. The RCA generated concatemers were subjected to library preparation and then sequenced on the nanopore. The concatemers (chain of long multiple copies) generated by RCA also creates repetitive genome regions. This increased read depth (repetitive sequence representations) allowed us to identify sites of true genetic polymorphisms, drastically discounting the fidelity concern with Nanopore. Consensus accuracy for 9.4 flowcell have been reported to be as high as Q40 (base call accuracy of 99.99%).

The long reads generated by the nanopore was analyzed using nanopore technologies' 'What's In My Pot' workflow. The workflow uses Centrifuge (Kim et al 2016) for taxonomic identification of viral species. Briefly, concatemers generated by RCA were sequenced on the nanopore and then taxonomically classified using Centrifuge.

3) The following sentence is hard to follow: “Classified reads for each TTV were used with Canu (v1.9, parameters:genomeSize=3k and -nanopore-raw) to assemble species-specific viral genomes. correspondingReference anelloviridae sequences obtained from the ICTV database (<https://talk.ictvonline.org/>).”

We have now reworded the text to read as following. “Reads were classified for each TTV, then *de novo* assembled with Canu (v1.9)to assemble species-specific viral genomes. Parameters

were set to assume a viral genome size of 3 kilobases (genomeSize=3kb) and nanopore error rate (--nanopore-raw).” This is now found on page 11,line 23 and page 23, line 1.

4) Some samples yielded up to four anelloviruses. Are these different species of anelloviruses? Were the nanopore sequence reads mapped against the ICTV species to count them? Was ORF1 used or any part of the genome? Was there a minimum % of sequence difference used to count different viruses or species?

This is an excellent observation. Sequences were taxonomically classified using Centrifuge, a microbial classification engine, that is part of ‘What’s in my Pot’ workflow used for real-time analysis of nanopore sequences. Centrifuge classifies based on Ferragini-Manzini (FM) index and searches both the read and corresponding reverse complement for short exact matches. The search begins at 16-bp and extends the match as far as possible. Classification of each read is based on the exact matches identified both in the read and the reverse complement with at least a complete 22-bp match. The sequences were mapped against Centrifuge’s database created from sequences in NCBI RefSeq (Pruitt et al 2005).

To confirm the presence of multiple anellovirus species, Sanger sequencing was done to validate the presence of four distinct anellovirus species identified in a single specimen. For this, reads from the sample that aligned to reference ICTV species for TTV7, TTV8, TTV9, and TTV18 were indexed and visualized on Integrated Genomic Viewer (IGV). These reads were used to design PCR primers targeting a 500bp region of ORF1 for each of the four TTV species to confirm the presence of the viral sequences. The ORF-1 region was targeted since based on ICTV guidelines

a cut-off value of 35% nucleotide sequence identity in the ORF-1 is used as a demarcation criterion for species identification.

5) What was done with the alpha, beta, gamma PCR products? How big were these anellovirus amplicons?

The nested PCR using genus specific primers was used to identify and differentiate alpha-, beta, and gamma infection. The results were used to determine the dynamics and abundance of anellovirus infections in PWID. The size of the resulting amplicons was used to differentiate the three genera – alpha- (112-117 bp), beta- (70-72 bp), and gamma- (88 bp) torquevirus.

6) Were infections with multiple pegivirus C ever detected?

All PCR identified pegivirus C infection were confirmed by sequencing the 671bp amplicon using Sanger sequencing with both forward and reverse primers. We did not observe multiple pegivirus infections that would have been inferred from heterozygous (double) peaks in the sequencing electropherogram.

7) PCRs were used for the two pegiviruses but not clear if the pegivirus-H used a nested or hemi-nested PCR (ie primers by Collins et al4).

We apologize for not being clear and now explain it in more detail in the text (Page 10, Lines 3-4). The PCR used for pegivirus H detection was done using a single round PCR that targeted a 160bp region in the NS2-3 junction. PCR amplification was done on cDNA synthesized using Superscript IV.

8) Was bleed over, the phenomenon observed with Illumina sequencing whereby a small fraction of reads from one sample end up with barcodes from another sample considered?

Does it occur with nanopore?

Like other next generation sequencing platforms bleed over is a possibility on nanopore platforms as well. However, they constitute a small percentage of read output and contribute to less than 0.5% of reads (Wick et al *PLoS Comput Biol* 2018).

9) The part of the material and methods, describing how viruses were counted, needs clarification.

Yes, thank you for encouraging us to be more clear. We have now clarified this in the materials and method.

2. Phylogenetic analyses.

1) Supp fig 2. The two trees are hard to follow and could benefit from more brackets or arrows to point out the relevant taxa. The HCV black taxa are clearly closely related but are readers supposed to interpret the other red circles that are closely linked as transmission pairs? Are these known pairs (or triplet in one case) confirmed by the PWIDs actually sharing injection equipment? Is the black HCV pair interesting because that relation between PWIDs was not known? Is the (weaker) linkage of the black pegiviruses used to confirm that these two PWID were directly connected (rather than there being an unknown intermediate PWID)? Non Baltimore pegi-C viruses are also clearly related. Are these longitudinal samples? From known contacts? Not clear what the point of all these taxa is? Is the point that there are

also other pairs of very closely related pegiviruses among the non-Baltimore subjects? These trees need to be better explained and labeled.

We agree completely and have modified supp fig 2 as suggested by the reviewer. The purpose of the figure is to show that pegivirus C sequences can identify not only drug use pairs previously identified by HCV sequences but also identify drug use pairs not observed by phylogenetic analysis of only HCV sequences. In other words, that in principle, they ‘add value’ to understanding transmission networks. We realize that if the viruses are spread by drug use that it stands to reason that their sequences might reveal when sharing occurred. However, while logical, we wanted to provide ‘proof of principle’ examples. The pair (black taxa) observed using HCV sequences were known to share injection equipment and was included as a positive control for analysis of pegivirus C sequence. While pegivirus C sequences confirmed the earlier observation (positive control was positive, black taxa), the sequences also identified other drug use pairs in the Baltimore cohort (the “experimental” or “unknown” pair). The previously unrecognized drug use pair were not known to share injection equipment but were in the same city, making it likely that this example shows in principle that the nonpathogenic virus sequences can reveal new information. The sequences from another city were included to serve as negative controls since identification of drug use pairs between the different geographical regions is highly unlikely. We have now clearly labelled drug use pairs identified using HCV and pegivirus C sequences to convey this message clearly. No drug sharing information was available for the non-Baltimore sequences. In addition, better labelling is used to denote reference genotype sequences and non-Baltimore sequences (Thailand). Thank you for encouraging us to make this more understandable.

2) The smaller TTV and HCV trees are also hard to comprehend at a glance. Is it the contention that TTV A and B are from a recent common ancestor (ie linked)? That subject B, C, and possibly D are also connected? That all the HCV taxa are closely connected? The very short length of virus sequenced (111 nucleotides for TTV and 337 for HCV) typically make such phylogenetic linkage rather weak. Full genome sequencing usually used for such transmission linkages.

Again, thank you for encouraging us to make this clearer. We have revamped and revised Figure 2. The simpler revised figure reveals sequence divergence represented as number of amino acid substitutions per site between TTV pairs identified in the study participants. Only TTV sequences that were classified by Centrifuge and confirmed as being present in the participants plasma using Sanger sequencing were included for the analysis. The amino acid sequences, obtained by Sanger sequencing, cover positions 275 to 400 of TTV-1 ORF. This sequence contains a relatively conserved region of ORF-1 in the 5' end followed by the first 100 amino acids of the HVR-1 region located in ORF-1 (Arze et al *Cell Host Microbe* 2021).

For this study phylogenetic analysis of HCV was based on the core-E1 region of the genome that Jens Bukh first described and donated to our lab. As with the TTV primers, the design begins in a conserved region of core and then extends into a more variable region of the envelope.

Analysis of this HCV sub-genomic region has been shown to contain sufficient phylogenetic information for transmission linkage studies (Simmonds et al *J Gen Virol* 1994, Ray et *J Infect Dis.* 2000, Rose et al *Infect Genet Evol.* 2019). While similar studies have not been carried with anelloviruses, we demonstrate the following using TTV sub-genomic region sequenced from 6 Baltimore sequences and 4 non-Baltimore sequences i) ability to differentiate intra and intra

subject TTV sequence variation and ii) lower amino acid sequence divergence between known drug use pairs in comparison to other pairs from within and outside the geographic region.

3) In fig 1 and S1 the pegiviruses cannot be differentiated from the anelloviruses. Are anelloviruses more likely than pegiviruses to be predictor of future HCV infection? Can anelloviruses alone do the job?

Great question. A higher prevalence of anellovirus among the study participants suggests they could serve as better predictors. However, more work needs to be done to understand their biology. For example, while we have observed an increase in pegivirus C seroprevalence in long term drug users, detection of pegivirus C RNA is associated with fewer years of drug use. (Thomas DL, *J Infect Dis.* 1997). Hence, while pegivirus C sequences will be helpful in identification of young drug users it might not be useful among long term drug users.

4) Also changing the term orphan viruses should be considered as it is not well known and a bit outdated. I suggest commensal viruses, non-pathogenic virome, or simply anelloviruses and pegiviruses. In text pegivirus C should be named as such not as Pegi-C.

We debated this term even prior to our first submission. Members of our team who agree with you are now delighted you have taken their side and we shift to your suggested term “non-pathogenic” virome and pegivirus C in the manuscript.

Reviewer #2 (Remarks to the Author):

This manuscript by Kandathil et al tried to address the virome in plasma in persons with

illegal substance abuse and connect that to the transmission of important pathogenic viruses including HCV. This work in its current form is not yet adequate to address important medical or epidemiological questions, and also the content is short from substantially interesting/convincing for nature communications.

The hypothesis on virome difference between persons with drug usage and non-users is correct yet not totally unexpected, shared needled as well as other paths surely will increase the amount and diversity of viruses in drug users, plus the immunological side effects of many illegal substances. Will these changes lead to more susceptibility to TTV or HCV? Maybe, but it's not tested in this manuscript. Plus, what about direct transfer of pathogenic viruses? In addition, it is quite unusual to see a manuscript NC without sections and only with two figures, this rather suggests insufficient work has been done on the research topic.

We thank the reviewer for the comments. We apologize for the formatting issues. The prior version was transferred from *Nature Medicine* and was formatted for their Brief Report section. The resubmitted manuscript now has been considerably edited and revised to fit the *Nature Communications* format. Thank you. We also agree that some have raised the question of immunological effects of opioids and even some in vitro evidence of different T cell biology. Nonetheless, we are not aware that work has ever been linked to 'susceptibility'. Instead, the net risk of transmission of HIV and HCV is strongly predicted by the drug use practices themselves (sharing needles, cookers, and other equipment and shooting in galleries). Thus, while we agree the question about increased susceptibility to TTV or HCV due to a weakened immune system is interesting, it is not a hypothesis we set out to address. Rather, this work was focused on the utility of non-pathogenic virome sequences to identifying drug use sharing

prior to acquisition of blood borne infectious diseases and to help strengthen viral sequence-based transmission studies that are currently based on phylogenetic analysis of the acquired viral pathogen(s).

Reviewers' Comments:

Reviewer #1:

Remarks to the Author:

The raw nanopore data files should be loaded in the SRA rather than available on request.

Fig 1 should differentiate the pegiviruses from the anelloviruses (maybe use circles for pegiviruses)?

The distribution of the 3 different anellovirus genera is not shown. All anellovirus lumped into one bin. Maybe add a table to get a better idea about the constitution of these commensal viromes.

Still not very clear in text how anelloviruses were counted. Based on presence/absence of different size bands the presence of alpha, beta, gamma genera and pegivirus can be deduced but not the number of different species or even variants within the same species. If only the presence/absence of PCR bands were used in fig 1 (likely since max of 4 -including pegivirus) were found) please state clearly. Also state that this an undercount as nanopore sequencing showed multiple anellovirus species within a single amplicon.

Maybe discuss that the conclusion are based on enumerating anellovirus genera and pegivirus C only and that full analyses of these anellovirus amplicons would have revealed a great number of species within each genus.

The genetics section is rather hard to follow.

Fig 2 Know drug pair and within host divergence is clear but not "unrecognized drug use pair " or "other Baltimore and non-Baltimore pairs".

Are unrecognized drug use pairs recognized because they carry closely related viruses? Did PWID admit to the relationship shown by sequencing? Are "other Baltimore and non-Baltimore pairs" closely related pairs of anellovirus sequences from PWID that could not possibly be sharing needles directly or indirectly? This figure is rather confusing. All the values in fig 2 are very low. Does it show all the pair-wise distances between members of the same species? What about simply plotting ALL the pair-wise distances when the same species of anellovirus are found in different subjects and highlighting those known contacts (possibly label those from the same city?).

"We then designed primers for partial amplification of the HVR-1 region located in ORF-1 of TTV". Please use alpha torquevirus instead of TTV. How come only 7 samples in 4 PWID relationships had their anellovirus amplicon sequenced (instead of 8)? Then it is written: "Using six sequences from five study participants and four non-Baltimore sequences that included ICTV reference sequences, amino acid sequence divergence was calculated using Mega X18 (Figure 2)". This seems a rather arbitrary selection of a subset of the amplicons? Were all alpha torquevirus amplicon sequenced and only 5 patients yielded ORF1 data? Fig 2 needs clearer explanations. This seems to have been done on only a subset of samples.

"By baseline, antibodies to pegivirus C were already detected in 35% and 30% among HCVneg to pos PWID and HCVneg to neg PWID, respectively. Accounting for resolved pegivirus C infection at baseline, further confirmed higher viral infections prior to HCV seroconversion in the HCVneg to pos PWID group compared to the HCVneg to neg PWID group (P=0.0041)." It is not clear how 35% versus 30% pegivirus C antibody detection results in a p value of 0.0041? Presumably here are more antibody test results going into this calculation that HCV neg to pos have higher incidence or prevalence of pegivirus C antibody detection?

Reviewer #1 (Remarks to the Author):

1) The raw nanopore data files should be loaded in the SRA rather than available on request.

The raw nanopore data has been available in SRA (Accession ID: PRJNA764703).

2) Fig 1 should differentiate the pegiviruses from the anelloviruses (maybe use circles for pegiviruses)?

We agree that information is important. We now provide it in even greater detail in an additional supplemental table (S. Table 1). This new table also helps us adopt your next suggestion. We opted not to complicate the message of Fig 1 any further since we had made a strong effort for clarity with our main figures in the prior revision.

3)The distribution of the 3 different anellovirus genera is not shown. All anellovirus lumped into one bin. Maybe add a table to get a better idea about the constitution of these commensal viromes.

Agreed. As suggested, we have now made an additional table (Suppl. Table 1) to show the distribution of the non-pathogenic virome components in detail.

4) Still not very clear in text how anelloviruses were counted. Based on presence/absence of different size bands the presence of alpha, beta, gamma genera and pegivirus can be deduced but not the number of different species or even variants within the same species. If only the

presence/absence of PCR bands were used in fig 1 (likely since max of 4 -including pegivirus) were found) please state clearly. Also state that this an undercount as nanopore sequencing showed multiple anellovirus species within a single amplicon. Maybe discuss that the conclusion are based on enumerating anellovirus genera and pegivirus C only and that full analyses of these anellovirus amplicons would have revealed a great number of species within each genus.

We apologize for the lack of clarity. Figure 1 is based on the presence/absence of PCR bands, and the process is now specifically mentioned in the text (Page 3, lines 64-65). As suggested, additional lines have also added in the discussion (page 7, lines 151-154)

5) The genetics section is rather hard to follow. Fig 2 Know drug pair and within host divergence is clear but not “unrecognized drug use pair “or “other Baltimore and non-Baltimore pairs”.Are unrecognized drug use pairs recognized because they carry closely related viruses? Did PWID admit to the relationship shown by sequencing?

We apologize for the lack of clarity. Let’s start with the last question. The insensitivity of PWID’s self-identification of relationships is the very reason we are asking if virus sequences might contribute new information. So, the answer is that those we referred to as ‘unrecognized’ were NOT admitted by the PWID. Moreover, they were not ‘recognized’ by the HCV sequencing. Those were first identified based on the pairwise distance of sequences. For those reasons we called them ‘unrecognized’. However, to reduce confusion, we will change it to ‘newly recognized’.

6) Are “other Baltimore and non-Baltimore pairs” closely related pairs of anellovirus sequences from PWID that could not possibly be sharing needles directly or indirectly? This figure is rather confusing. All the values in fig 2 are very low. Does it show all the pair-wise distances between members of the same species? What about simply plotting ALL the pair-wise distances when the same species of anellovirus are found in different subjects and highlighting those known contacts (possibly label those from the same city?).

Thank you for helping us better explain and represent the data. For figure 2, sequences from the same species (TTV7 and TTV8) were used to determine the pair wise distance so that we would not have an apples/oranges type problem with the scale. We decided on TTV7 and TTV8 since they had the highest prevalence in the PWID cohort based on taxonomic classification of the nanopore reads. Fortunately, since then and in agreement with our decision is a recent taxonomic reassignment of TTV8 as TTV7 (Varsani et al 2021). We have now differentiated pairwise distances between other Baltimore PWID pairs and all other pairs. All other pairs refer to sequence distance between Baltimore PWID and ICTV/GenBank sequences.

7) “We then designed primers for partial amplification of the HVR-1 region located in ORF-1 of TTV”. Please use alpha torquivirus instead of TTV. How come only 7 samples in 4 PWID relationships had their anellovirus amplicon sequenced (instead of 8)?

Thank you for the suggestion. We have now changed TTV to alphatorquevirus. Among the transmission pairs, one participant indicated sharing with two different participants,

independently. Hence, because there was a common subject in two PWID pairs, sequencing of seven samples was necessary, instead of eight.

8) Then it is written: “Using six sequences from five study participants and four non-Baltimore sequences that included ICTV reference sequences, amino acid sequence divergence was calculated using Mega X18 (Figure 2)”. This seems a rather arbitrary selection of a subset of the amplicons? Were all alpha torquevirus amplicon sequenced and only 5 patients yielded ORF1 data? Fig 2 needs clearer explanations. This seems to have been done on only a subset of samples.

We apologize for not being clear on the sample selection. Among the seven study participants that were sequenced using nanopore, two study participants did not have sufficient read coverage to design primers to cover ORF-1 amino acid positions 275-400. Hence, Sanger sequencing was done only on 5 participants. We then included the reference ICTV sequences and other highly similar full-length sequences to the ICTV sequences.

9) “By baseline, antibodies to pegivirus C were already detected in 35% and 30% among HCVneg to pos PWID and HCVneg to neg PWID, respectively. Accounting for resolved pegivirus C infection at baseline, further confirmed higher viral infections prior to HCV seroconversion in the HCVneg to pos PWID group compared to the HCVneg to neg PWID group (P=0.0041).” It is not clear how 35% versus 30% pegivirus C antibody detection results in a p value of 0.0041? Presumably here are more antibody test results going into this

calculation that HCV neg to pos have higher incidence or prevalence of pegivirus C antibody detection?

We apologize for not being clear on how the p value was calculated. After initially assessing non-pathogenic virome components using PCR, we reasoned that resolved pegivirus infections might be underestimating the ‘true’ number of infections in both the PWID groups and controls. Hence, we determined pegivirus-C and pegivirus-H seroprevalence in the study samples and counted seropositivity as another infection. For example, if someone had two viruses in blood and was seropositive, that was counted as a total of 3. Thus, even though 35% is not much higher than 30%, the serological information further reduced the probability of the null hypothesis from 0.0031 (calculated when considering only viremia) even lower to 0.0041 (Figure 1). The latter represents the sum of the serologic and DNA/RNA data. This process mirrors how new information is meta-analyzed/integrated into the initial hypothesis testing data, not as a stand-alone study but rather to support/not support the existing results.

Figure 1: Greater accumulation of non-pathogenic virome in PWID who later acquire HCV (HCV_{neg to pos}) than those who do not (HCV_{neg to neg}) and controls who do not inject drugs.

Greater non-pathogenic virome components were observed in HCV_{neg to pos} PWID (n=20) when compared to HCV_{neg to neg} PWID (n=20) and non-injection drug users (controls) at baseline when assessing viremia using PCR infection and b) also when considering both past and present infections with pegiviruses. Error bar denotes median. Two tailed P value were calculated using Mann Whitney or Kruskal-Wallis test. HCV was not included in the analysis.

Reviewers' Comments:

Reviewer #1:

Remarks to the Author:

Happy with revised manuscript.